# Improvements in Quality Control and Library Preparation for Targeted Sequencing Allowed Detection of Potentially Pathogenic Alterations in Circulating Cell-Free DNA Derived from Plasma of Brain Tumor Patients

**DOI:** 10.3390/cancers14163902

**Published:** 2022-08-12

**Authors:** Paulina Szadkowska, Adria-Jaume Roura, Bartosz Wojtas, Kamil Wojnicki, Sabina Licholai, Tomasz Waller, Tomasz Gubala, Kacper Zukowski, Michal Karpeta, Kinga Wilkus, Wojciech Kaspera, Sergiusz Nawrocki, Bozena Kaminska

**Affiliations:** 1Nencki Institute of Experimental Biology, 02-093 Warsaw, Poland; 2Postgraduate School of Molecular Medicine, Medical University of Warsaw, 02-007 Warsaw, Poland; 3Academic Computer Centre CYFRONET AGH, 30-950 Krakow, Poland; 4Sano Centre for Computational Medicine, 30-054 Krakow, Poland; 5Regional Science and Technology Centre, 26-060 Podzamcze, Poland; 6Department of Neurosurgery, Medical University of Silesia, Regional Hospital, 41-200 Sosnowiec, Poland; 7Department of Oncology, University of Warmia and Mazury in Olsztyn, 10-228 Olsztyn, Poland

**Keywords:** liquid biopsy, glioma, circulating tumor DNA, ccfDNA, targeted NGS

## Abstract

**Simple Summary:**

Malignant brain tumors present an enormous challenge due to their genetic heterogeneity, and the difficulties in accessing them impede a precise diagnosis. Numerous genetic alterations have been described, and some of them can assist in personalized therapy for glioma patients. Brain biopsy is an invasive procedure with potentially deleterious complications. Liquid biopsy from a patient’s plasma may provide a less invasive method for diagnosis and personalized therapy selection. We performed targeted next-generation sequencing of tumors and circulating cell-free DNA (ccfDNA) from 84 brain tumor patients. We detected tumor-specific genetic alterations in ccfDNA in 5 out of 80 glioma patients and potentially pathogenic alterations in ccfDNA from the plasma of 29 out of 80 glioma patients. Despite a low efficacy, with further improvements, the detection of genetic alterations in ccfDNA holds promise for noninvasive diagnosis, which may revolutionize personalized therapy for these deadly tumors.

**Abstract:**

Malignant gliomas are the most frequent primary brain tumors in adults. They are genetically heterogenous and invariably recur due to incomplete surgery and therapy resistance. Circulating tumor DNA (ctDNA) is a component of circulating cell-free DNA (ccfDNA) and represents genetic material that originates from the primary tumor or metastasis. Brain tumors are frequently located in the eloquent brain regions, which makes biopsy difficult or impossible due to severe postoperative complications. The analysis of ccfDNA from a patient’s blood presents a plausible and noninvasive alternative. In this study, freshly frozen tumors and corresponding blood samples were collected from 84 brain tumor patients and analyzed by targeted next-generation sequencing (NGS). The cohort included 80 glioma patients, 2 metastatic cancer patients, and 2 primary CNS lymphoma (PCNSL) patients. We compared the pattern of genetic alterations in the tumor DNA (tDNA) with that of ccfDNA. The implemented technical improvements in quality control and library preparation allowed for the detection of ctDNA in 8 out of 84 patients, including 5 out of 80 glioma patients. In 32 out of 84 patients, we found potentially pathogenic genetic alterations in ccfDNA that were not detectable in tDNA. While sequencing ccfDNA from plasma has a low efficacy as a diagnostic tool for glioma patients, we concluded that further improvements in sample processing and library preparation can make liquid biopsy a valuable diagnostic tool for glioma patients.

## 1. Introduction

Recent advances in the detection and characterization of ctDNA have allowed the implementation of liquid biopsy in clinical practice. Several single- or multi-gene assays for detecting genetic alterations in ccfDNA are used for better cancer diagnosis and molecularly targeted therapy recommendations. A few assays have been approved by the American Federation of Drug Administration [1,2]. ccfDNA originates from degraded DNA fragments that are released into the blood and all other bodily fluids due to ongoing apoptosis, necrosis, or secretion from either normal or malignant cells [3,4]. Released fragments of ccfDNA and the presence of ctDNA can be assessed using many different methods: qPCR [3,5], droplet digital PCR (ddPCR) [4], atomic force microscopy (AFM) [5], massive parallel sequencing (MPS) [6], microchip-based capillary electrophoresis [7], or agarose gel electrophoresis [8]. Elevated levels of ccfDNA are detected in many pathological conditions: advanced cancer, sepsis, myocardial infarction, physical trauma, pregnancy, and transplant graft rejection [9,10,11]. ccfDNA circulates as fragments ranging in length from 120 to 220 base pairs (bp), with the peak at 167 bp, but dimers and trimers of this length have been found [10]. The estimated half-life of ccfDNA in circulating blood varies from 2 min to 2 h [12]. This rapid turnover allows for a snapshot of the tumor’s mutational landscape, which provides valuable diagnostic information. ccfDNA contains nuclear and mitochondrial DNA, which impacts its structure and stability [13]. ctDNA can be used as a marker for detecting cancer-specific alterations or tracking tumor evolution [14,15,16].

Malignant gliomas are the most frequent primary brain tumors in adults [17] and are classified by the World Health Organization (WHO) as either grade 3 or 4 gliomas. The most aggressive is glioblastoma (GBM, G4), which is characterized by highly infiltrative growth, multiple genetic alterations, and high resistance to therapy, resulting in rapid recurrence and a high mortality rate. The median overall survival of GBM patients is 14.5 months from the time of diagnosis, despite extensive surgical resection, radiotherapy, and chemotherapy [18]. Multiplatform genomic, epigenetic, and proteomic analyses of these tumors by The Cancer Genome Atlas (TCGA) consortium [19] revealed predominant, recurrent alterations, molecular subtypes, and potential therapeutic clues [19,20,21,22]. Recurrent somatic alterations in genes such as *TP53*, *PTEN*, *NF1*, *ATRX*, *EGFR*, *PDGFRA*, and *IDH1* have been reported [20,23]. Despite many developments in oncology, there has been no progress in therapy for malignant gliomas. Due to the shortage of efficient treatment, the detection of rare but targetable genetic changes is of interest, which raises the possibility for tailored therapy for selected patients. This is exemplified by the inhibitors of the mutated BRAF V600E targeting the alteration present in 1% of glioma patients [24]; inhibitors of Aurora kinases (AURK), which are upregulated in GBMs; or AURK inhibitors, which have a synergistic or sensitizing effect when combined with standard therapy [25,26].

The biopsy of brain tumors is invasive and risky, particularly in elderly patients. Moreover, sampling is highly biased due to cellular and genetic tumor heterogeneity. Some brain tumors are in the eloquent brain regions, which makes biopsy difficult and likely to cause postoperative complications. It is essential to provide real-time quantitative information regarding the tumor burden and qualitative information on genetic profiles that can be used for better diagnosis, prognosis, and outcome prediction. The cerebrospinal fluid (CSF) is in direct contact with tumors and may serve as a better source of ctDNA. CSF-derived ccfDNA has been used to characterize genomic alterations, the dynamics of tumor growth, and genomic evolution. Several studies have reported the presence of ctDNA in the CSF of patients with primary brain tumors or metastatic lesions [2,27,28,29]. The quality and quantity of ccfDNA, which is typically isolated from the plasma of glioma patients, are low; standard methods, effective in other tumors, produce a poor outcome. We hypothesized that improvements in various steps of ccfDNA isolation, library preparation, and sequencing may reinforce the quality of results.

In the present study, we isolated matching samples of tDNA, ccfDNA, and whole-blood reference DNA (gDNA) from 84 brain tumor patients and analyzed them by targeted next-generation sequencing (NGS). The cohort included 80 glioma patients, 2 metastatic cancer patients, and 2 primary CNS lymphoma (PCNSL) patients. We compared the pattern of genetic alterations in tDNA with that of ccfDNA and gDNA. Owing to technical improvements allowing for precise quality and quantity control and application of targeted NGS, ctDNA was detected in 8 out of 84 patients, including 5 out of 80 glioma patients. Some ccfDNA showed somatic alterations that were not detectable in the matching tDNA. While the sequencing of ccfDNA from plasma has low efficacy in the case of brain tumor patients, which prevents the use of this method as a diagnostic tool now, we concluded that further improvements to the isolation, processing, and sequencing of ccfDNA might make liquid biopsy available for glioma patients in the future.

## 2. Materials and Methods

### 2.1. Patients

Freshly frozen tumors and corresponding blood samples were collected from brain tumor patients. The blood samples were collected before and after surgery. Detailed descriptions of the patient cohort are presented in Appendix A. Each patient gave written consent for the use of their blood and tumor tissues. All the procedures that involved human participants were performed in accordance with the institutional ethical standards and were approved by the ethics committee of the Medical University of Silesia (KNW/0022/KB1/2/I/17). In the NGS analysis, we used presurgery plasma-derived ccfDNA collected from 84 patients: 80 patients with WHO G3 and G4 gliomas, 2 patients with primary central nervous system lymphoma (PCNSL), and 2 patients with anaplastic thyroid cancer metastasis and adenocarcinoma lung metastasis. Copy number alteration targeted sequencing was additionally performed on 4 ccfDNA samples that displayed the ctDNA signal in the primary analysis.

### 2.2. DNA Isolation

tDNA was extracted from freshly frozen (−80 °C) tumor tissue samples using Trizol reagent (Thermo Fisher Scientific, Waltham, MA, USA), following the manufacturer’s protocol. gDNA was isolated from whole blood samples that were stored and frozen (−20 °C) in EDTA-coated tubes prior to isolation, using a QIAamp DNA Blood Mini Kit (Qiagen, Hilden, Germany), following the manufacturer’s protocol. The blood collected for ccfDNA isolation was stored in ccfDNA PAXgene tubes (PreAnalytiX, Homberchtikton, Switzerland) prior to isolation. The postsurgical blood used for ccfDNA isolation was collected from most patients 2–3 days after surgery, and 4–5 days postoperation in some cases. Presurgical blood samples were collected up to 24 h prior to surgery. In accordance with the ccfDNA *PAX*gene (PreAnalytiX, Homberchtikton, Switzerland) manufacturer’s recommendations, the blood samples were stored at room temperature for up to 10 days. The blood tubes (PreAnalytiX, Homberchtikton, Switzerland) were centrifuged at room temperature (15–25 °C) for 15 min at 1900× *g*. The obtained plasma was transferred into a 15 mL conical-bottom centrifugation tube without disturbance of the buffy coat. The cellular fraction was centrifuged for 10 min at room temperature (15–25 °C) at 1900× *g* for further purification of the plasma. ccfDNA was isolated from the obtained plasma using a QIAamp Circulating Nucleic Acid Kit and QIAvac system (Qiagen, Hilden, Germany). The isolation was performed according to the manufacturer’s protocol, with appropriate amounts of reagents selected depending on the volume of the input material. The protocol assumes a volume of input material of 1–5 mL of plasma. The obtained ccfDNA was stored at −80 °C until further processing. For some patients, additional whole-blood reference samples were not available; in this case, after plasma removal for ccfDNA isolation, the remaining fraction was stored at −20 °C and later used for gDNA isolation using a QIAamp DNA Blood Mini Kit (Qiagen, Hilden, Germany), so in this case, both ccfDNA and gDNA were isolated from the same blood sample.

### 2.3. Design of Targeted Glioma-Related Gene Enrichment Panel

To capture a wide spectrum of somatic mutations, tDNA and gDNA were sequenced using a broad 664-gene panel (Appendix A). A SeqCap EZ Custom Enrichment Kit was used, which is an exome enrichment design that targets the latest genomic annotation GRCh38/hg38. The vast majority of the genes (578) were selected from a Roche Nimblegen Cancer Comprehensive Panel (based on the Cancer Gene Consensus from the Sanger Institute and NCBI Gene Tests). Eighty-six epigenetics-related genes were additionally included (genes coding for histone acetylases and deacetylases, histone methylases and demethylases, DNA methylases and demethylases, and chromatin-remodeling proteins) based on a literature review [30,31,32]. The targeted deep sequencing of ccfDNA was performed using a narrow 50-gene Sure Select XT HS custom panel (covering 411,483 kbp) that targets the genomic annotation, GRCh19/hg19 (Design ID: 3216011). All selected gene regions were included in the 664-gene larger panel (Appendix A). For the 50-gene panel, we selected the genes most frequently mutated in the previously analyzed patient cohort of 182 glioma samples [33]. The copy number alteration of Sure Select XT HS custom probes covering the same 50-gene region was additionally tested (Design ID: A3224001, Appendix A).

### 2.4. Sequencing tDNA and gDNA

The libraries of tDNA and gDNA were prepared using a KAPA HyperPlus Kit, according to the SeqCap EZ HyperCap Workflow user’s guide (version 2.3). The library preparation used 100 ng of DNA. After the enzymatic fragmentation of the material to obtain 180–220 bp DNA fragments, the end repair and A-tailing were performed. Next, the indexed adapters were ligated, the double-size selection was performed, and the libraries were amplified. The concentration of the resulting libraries was determined by a Quantus Fluorometer with a QuantiFluor ONE Double-Stranded DNA System (Promega, Madison, WI, USA), and the quality check was performed using an Agilent Bioanalyzer (Agilent Technologies, Santa Clara, CA, USA). The obtained libraries were mixed in equimolar concentrations to form a 1400 ng pool. After COT (Human Cot-1 DNA^®^, NimbleGen SeqCap EZ Accessory Kit v2, Roche, Basel, Switzerland) and complementary adapter oligos (SeqCap Adapter Kit Band hybridization, Roche) were added, the sample was condensed using a PCR clean speed vac for 30 min at 60 °C. The resulting pool was mixed with probes and additional reagents from a SeqCap EZ Custom Enrichment Kit, denatured at 95 °C for 10 min and incubated at 47 °C for at least 17 h to allow proper probe binding. After overnight incubation, the mixture of pooled libraries was purified using special HyperCap beads (HyperCap Bead Kit, Roche) and later amplified. During that step, the libraries were enriched with fragments of interest. The quality of the obtained libraries was evaluated using an Agilent Bioanalyzer with a High-Sensitivity DNA Kit (Agilent Technologies, Palo Alto, CA, USA). The quantification of the libraries was performed using a Quantus Fluorometer and a QuantiFluor Double-Stranded DNA System (Promega, Madison, WI, USA). The libraries were run in a rapid-run flow cell and paired-end sequenced (2 × 76 bp) on a HiSeq 1500 (Illumina, San Diego, CA, USA).

### 2.5. ccfDNA Sequencing

The quality of ccfDNA was evaluated using an Agilent Bioanalyzer with a High-Sensitivity DNA Kit (Agilent Technologies, Palo Alto, CA, USA). Most of the samples contained fragments that averaged around 167 bp. Samples that only displayed fragments longer than 2000 bp or had no detectable nucleic acid signal were eliminated from further processing. The libraries were prepared using a Sure Select XT HS Target Enrichment System for Illumina Paired-End Multiplexed Sequencing Library (Agilent Technologies, Palo Alto, CA, USA) according to the manufacturer’s protocol (version C 2 July 2019), with only a few adjustments. The fragmentation step was skipped for all the ccfDNA samples because properly isolated DNA is characterized by a 120–200 bp fragment size.

The libraries were prepared from the scarce amounts of ccfDNA material, which varied between 0.5 and 10 ng depending on the sample. The ends were repaired and a dA-tail was added to the 3′ ends. The next step was the ligation of molecular-barcoded adapters. During this step, unique molecular identifiers were attached to each DNA fragment, labeling each as an original and unique sequence prior to PCR amplification. This step was crucial in allowing a clear verification of false positives during the later stages of the bioinformatic analysis of the sequenced data. The remaining molecular-barcoded adapters were removed by AM Pure Bead purification prior to the PCR-amplification step. Due to the small quantity of ccfDNA, 14 PCR cycles were performed, during which SureSelect XT HS Index Primers were added to label each sample.

According to current reports [34,35], the enrichment of ccfDNA in shorter fragments can improve tDNA detection, so an additional right-sided size-selection step, using AM Pure Beads (Beckman Coulter, Brea, CA, USA), was added to the protocol. The quality of the resulting libraries was evaluated using an Agilent Bioanalyzer with a High-Sensitivity DNA Kit (Agilent Technologies, Palo Alto, CA, USA). The hybridization and capture were conducted according to the manufacturer’s protocol (version C 2 July 2019). Finally, the quality of the obtained libraries was evaluated using an Agilent Bioanalyzer as described above. The libraries were run in a rapid-run flow cell and were paired-end sequenced (2 × 100 bp) on a HiSeq 1500 (Illumina, San Diego, CA, USA), as recommended by the manufacturer.

### 2.6. Bioinformatic Analyses

Somatic variants pipeline. The FASTQ files obtained from sequencing the ctDNA and gDNA samples were processed with a trimmomatic program [36] to remove low-quality reads and sequencing adapters. Filtered and trimmed reads were mapped to the human genome (hg38) by NextGenMap aligner (http://cibiv.github.io/NextGenMap/ (accessed on 8 November 2021)) [37]. Any read duplicates were marked and removed by Picard (https://broadinstitute.github.io/picard/ (accessed on 8 November 2021)) [38], and only properly oriented and uniquely mapped reads were considered for further analysis. For somatic ctDNA calls, a minimum coverage of 10 reads was established. Additionally, variants with strand-supporting-read bias were discarded. Only coding variants with damaging predicted SIFT values (>0.05) were selected. The ProcessSomatic method from VarScan2 [39] was applied to extract high-confidence somatic cells based on variant allele frequency and Fisher’s exact test *p*-value. The final subset of variants was annotated with Annovar (http://annovar.openbioinformatics.org/en/latest/ (accessed on 8 November 2021)) [40], using the latest databases versions (refGene, clinvar, cosmic, avsnp150 and dbnsfp30a). Finally, the maftools R library [41] was used to analyze the resulting somatic variants.

ccfDNA data analysis. In the current study, tumor and blood samples were processed using a dedicated pipeline based on open-source bioinformatics tools, while the ccfDNA samples were treated using SureCall (https://www.agilent.com/en/product/next-generation-sequencing/hybridization-based-next-generation-sequencing-ngs/ngs-software/surecall-232880 (accessed on 16 January 2022)), which is dedicated software provided by a library-preparation reagent manufacturer. The raw sequencing reads from both tumor and blood samples were converted to fastq files with bcl2fastq software (https://emea.support.illumina.com/sequencing/sequencing_software/bcl2fastq-conversion-software.html (accessed on 16 January 2022)) from Illumina. The quality control of obtained reads was performed using the FastQC tool. The raw reads obtained from the ccfDNA samples were processed and converted to a fastq format, which allowed the mutant variants to be detected using SureCall software. Each somatic mutation within the tumor was assigned to a corresponding read from ccfDNA. Annotation of variants obtained by ccfDNA sequencing was performed.

To find somatic variants, reads of sufficient quality were mapped to the human reference genome, hg19, using the bwa package and standard parameters because the library manufacturer’s pipeline is a proprietary software solution that could not be modified to be compatible with hg38. This was followed by a recalibration, de-duplication, and variant-calling in somatic mode using the appropriate tools from the GATK package [42]. Among others, BaseRecalibrator, MarkDuplicates, and Mutect2 were used. The obtained vcf files were annotated using the Annovar (http://annovar.openbioinformatics.org/en/latest/ (accessed on 16 January 2022)) [40] package with appropriate databases (refGene and ClinVar, among others). Mutations, which were supported for at least 10 raw reads and found only in tumor samples, were filtered out and treated as potentially pathogenic. Somatic variant calls were retained that presented at less than 1% mutant allelic frequency in the gDNA, but with at least 1% allelic frequency and at least 3 reads supporting variant alleles in the tumor samples. We filtered the mutations reported in dbSNP (v137) and the 1000 Genomes database. Copy number variations (CNVs) were detected using ADTEx (http://adtex.sourceforge.net (accessed on 15 February 2022)) with default parameters. The germline CNVs from each patient were identified using the blood sample and normal human HapMap DNA sample NA18535 (Coriell Institute) for each captured region (exonic region). The somatic CNVs were identified using paired blood DNA–tumor DNA samples for each exon. The data were deposited to European Genome-phenome Archive EGA (http://www.ebi.ac.uk/ega/ (accessed on 25 July 2022)), hosted by the European Bioinformatics Institute (EBI) under accession numbers EGAS00001006451 and EGAD00001009080.

Statistical analysis. The statistical significance was calculated using a *t*-test with GraphPad Prism v6 (GraphPad Software, San Diego, CA, USA). *p*-values < 0.05 were considered significant.

## 3. Results

### 3.1. Cohort Characteristics and Quality Control

The initial cohort included 126 patients from whom clinical data, including age, sex, and diagnosis, were collected. The majority of the cohort was composed of patients with gliomas: WHO grade 4 (92), grade 3 (11), grade 2 (15), and grade 1 (1). There were also PCNSL (2), metastatic cancers (4), brain aneurysm (1). The clinical characteristics of this cohort are available in Appendix A and as a summary in Figure 1A,B. The matching pairs of bulk-tumor and whole-blood samples were collected. The blood for ccfDNA isolation was collected from most patients prior to and after surgery.

### 3.2. Precise Quantification of ccfDNA Demonstrates Importance of Fast Blood Processing and Marked Increase in ccfDNA after Surgery

The determination of the purity, quality, and quantity of the starting material was required for the precise control of the further steps. We used microchip-based capillary electrophoresis to quantify those parameters. While this protocol is not regular, we found it produces reliable and precise results for the quantity and quality of low-abundance DNA.

Due to collecting materials from different surgery clinics, the blood for ccfDNA isolation was stored in PAXgene blood ccfDNA tubes for varying times prior to isolation, allowing a comparison of its quality after short- and long-term storage (Figure 2A). ctDNA is usually short in size [29,30]; therefore, to improve its detection, we performed right-sided size selection using magnetic beads. Some blood samples were not eligible for ccfDNA isolation due to hemolysis; others failed quality control. In summary, ccfDNA (ranging in size from 100 to 500 bp) was quantified for 95 patients (Figure 2B). The unpaired *t*-tests confirmed that ccfDNA isolated within 24 h after the blood collection (*n* = 8) had a significantly higher yield (Figure 2C) compared with the rest of the samples that were not subjected to ccfDNA isolation for more than 24 h (*n* = 87).

The pre- and postsurgery isolated ccfDNA concentrations were measured in 19 patients (Figure 3A), and a significant increase in ccfDNA levels in the blood after surgery was detected (Figure 3B). The library preparation of the complete sets of preoperative ccfDNA, ctDNA, and reference gDNA was successful in the case of 84 patients.

The libraries were prepared from various quantities (0.5–10 ng) of ccfDNA. Quality control step using Bioanalyzer (Agilent Technologies, Palo Alto, CA, USA) excluded samples with no visible ccfDNA (Figure 4A) or ones that had significant amounts of long DNA fragments polluting ccfDNA samples (Figure 4B). Samples with long DNA fragment pollution, but distinguishable ccfDNA signal were used in further library preparation (Figure 4C). Prior to hybridization, right-sided size selection was performed using AM Pure Beads (Beckman Coulter) to enrich the final library into shorter fragments (Figure 4D) and remove fragments longer than 500 bp. The libraries were prepared without DNA fragmentation. Such small improvements may increase the detection of ctDNA, according to recent reports [34,35].

### 3.3. Identification of Somatic and Germline Variants

First, we performed targeted sequencing to identify somatic and germline variants in the tDNA and gDNA samples. The most frequently found somatic mutations in the cohort were *PTEN*, *TP53*, *EGFR*, *ATRX*, *IDH1*, and *NF1* (Figure 5A), in coherence with the findings in our previous study [43]. We used an oncodriveCLUST algorithm [44] to identify cancer drivers based on mutational clustering and found several variants enriched at the *TP53* (five clusters), *RECQL4* (one cluster), PIK3CA (two clusters), and *IDH1* (one cluster) genes, among many others (Figure 5B). The variations between the tumor samples’ mutation penetration were reviewed by comparing the allele frequencies of detected somatic variants within specific gene regions (Figure 5C).

We found that *TP53* and *PTEN* frequently harbored mutations with high variant allele frequencies (VAFs), suggesting the presence of homozygous mutations in some patients (Figure 5C). The *RB1*, *EGFR* and *CDKN2A* genes exhibited mutations with VAFs close to 0.5, indicating the loss of heterozygosity in those mutations [45]. In terms of the germline analysis, we found a mutation in the *AKAP9* (T1334fs) gene in 36 of the 89 patients; this variant had no clinical or mutational annotation format (MAF) data. The germline variant allele frequencies were close to 0.5, implying heterozygous mutations.

### 3.4. Identification of Somatic Variants in ccfDNA

ccfDNA was sequenced using the preselected 50-gene custom panel to achieve a deep sequencing coverage. Based on a previous study [33], we selected the top 50 altered genes in the Polish population of 182 gliomas that had diagnostic or prognostic potential (Appendix A). The obtained data were compared with those of cells from the analysis of the somatic variants shown above. This resulted in finding the same somatic genetic alterations in both ccfDNA and ctDNA of eight patients, including five WHO grade 3 or 4 glioma patients, one1 PCNSL patient, and two metastatic brain patients (Table 1).

The *SMARCA4* mutation had a 0.71 allele frequency (AF) in the ctDNA, and a significant mutation penetration was detected at AF 0.23 in ccfDNA, but no change was registered in gDNA (patient ID 59). The applied pipeline was very stringent and might not detect all single-nucleotide variations (SNVs) present in ccfDNA. For example, the somatic variant in *IDH1* was detected using IGV genome browser [46] but was not detectable as an SNV from the Surecall pipeline.

### 3.5. Potentially Pathogenic Variants Found in ccfDNA, but Not gDNA

Genetic variant databases have recently improved, thus allowing identification of extremely rare variants (based on EXAC, TOPMED, gnomAD, and 1000 Genomes data projects). Pathogenic, likely pathogenic, or disease-coexisting variants are also registered in databases such as ClinVar or COSMIC. Malignant gliomas (particularly GBMs) are genetically heterogeneous [47]. Thus, removing a tumor fragment is intrinsically limited to encompassing its complete mutational heterogeneity. Based upon this assumption, an additional analysis was performed. First, the variants detected in ccfDNA but not in gDNA were filtered, then potentially pathogenic variants were identified. The COSMIC registered coding variants present in ccfDNA but not in gDNA are presented in Figure 6 and Table 2. Most of the selected SNVs were also registered in the ClinVar database as pathogenic or likely pathogenic; some were extremely rare in the population (MAF, AF 1000G, and gnomAD), as shown in Table 2. Altogether, we discovered potentially pathogenic variants in ccfDNA in 25 brain tumor patients.

The SNVs detected in ccfDNA but not in gDNA were filtered out and compared with the somatic variants identified in the previously analyzed cohorts. We found potentially cancer-originating mutations in the ccfDNA samples from 16 patients. The comparison of ccfDNA variants that were not detected in gDNA with somatic variants detected in another glioma cohort (*n* = 57) from our previous studies [33] is presented in Table 3 and compared with somatic variants from the current study in Table 4. This analysis yielded a common *PTEN* benign (carrier) mutation that was frequently detected in ccfDNA (10 patients) and likely pathogenic *TP53* and *EGFR* variants present in two additional samples of ccfDNA (Table 3). For example, the *EGFR* variant (rs149840192) that was found in the ccfDNA of patient 64 was registered in 36 brain tumor cases in the COSMIC database and was confirmed as somatic in one patient from our previous study [33] and in three patients from the current study (Table 4). The complete score of these analyses yielded a set of genetic variants found in ccfDNA but not in gDNA, which suggests they originated from the tumors (Figure 7).

### 3.6. Detection of Copy Number Alterations in ccfDNA

Copy number alteration (CNA) was another significant somatic alteration in gliomas which often showed a distinctive landscape with synchronous genomic gains or losses [48]. Recently, a new, interesting method has emerged that involves CNA analysis with the targeted panel sequencing. We tested its applicability to ccfDNA sequencing. Four samples of ccfDNA, in which we detected positive ctDNA signals, were chosen. Libraries were prepared from both ccfDNA and gDNA using a SureSelect XT library prep kit. Special probes that determine a copy-number change with a custom design covering the same gene region as the original SureSelect XT custom panel were used. We found numerous CNAs in ccfDNA. The amplifications that were registered in the COSMIC database are reported in Figure 8.

## 4. Discussion

The detection of ctDNA in brain tumors is still insufficient to incorporate plasma-derived liquid biopsy into clinical practice for glioma patients. CSF has been proposed as a better source of ccfDNA [14,15,27] as ctDNA is more abundant in CSF than in plasma, and the sequencing of ccfDNA isolated from CSF more comprehensively characterizes the genomic alterations of glioma. However, collecting a CSF sample by lumbar puncture is a highly invasive procedure and may cause additional complications in brain tumor patients. Therefore, we exploited a few technical improvements to find actionable genetic changes in ccfDNA from the blood plasma of malignant glioma and metastatic patients.

In the present study, we underlined how slight improvements in isolation, library preparation, and mutational analyses of ccfDNA might lead to better detection of tumor-specific genetic alterations. The results can be summarized as follows: (1) we established a reliable method to determine the precise quantity and quality of ccfDNA using an Agilent Bioanalyzer with a commercial High-Sensitivity DNA Kit; (2) we prepared libraries from ccfDNA without a fragmentation step and performed right-sided size selection, which improved the quality of library preparation; (3) using the 50-gene custom panel, we found somatic variants in ccfDNA of eight patients, which is consistent with those detected in the tumors; (4) we found several somatic variants in ccfDNA that are likely pathogenic but have not been detected in tDNA; (5) we implemented a protocol for the detection of copy number alterations with a commercially available library preparation kit and custom gene panel, which revealed copy number amplifications in ccfDNA that are likely pathogenic.

Our results suggest that there is substantial room for improvement in the sequencing of ccfDNA by shortening ccfDNA isolation time, size selection, and library preparation and deep sequencing with targeted panels. We confirmed that processing blood within 24 h after collection significantly increases the yield of isolated ccfDNA. Our findings agree with reports that showed improved ctDNA detection rates upon instantaneous plasma separation (within 2 h after blood collection) and freezing (at −80 °C) prior to ccfDNA isolation [16]. Measuring the concentration of a specific length of ccfDNA and checking the size distribution ratio using a Bioanalyzer with a commercial High-Sensitivity DNA chip allows the study of quantity, quality, and gDNA contamination levels, which can direct library preparation and ccfDNA isolation procedures. Small, targeted panels that include the most common, actionable mutations may facilitate personalized therapy for glioma patients. In our study, the applied pipeline of data analysis was very stringent, and we found cases where manually reviewing the BAM files using the IGV browser (Broad Institute, USA) showed alterations not identified by the mutect2 or Surecall pipelines. For example, a well-known *IDH1* gene substitution in ccfDNA was detected in the ccfDNA BAM file in 5 out of a total of 1211 reads in the IGV browser but not in the results generated using the mutect2 or Surecall pipelines. The variant was lost in data processing, and it was likely removed by software quality control.

The most interesting finding was the nongermline, pathogenic variants in ccfDNA that were not detected in the matched tDNA. After scrutinizing numerous public databases and previous datasets of almost 280 gliomas sequenced with the custom gene panel (664 cancer-related genes), we are confident that the discovered SNVs are pathogenic. Detecting these alterations in ccfDNA, but not in the matching tumor DNA, may be explained by the reported genetic heterogeneity of glioblastoma and regional sampling of a tumor acquired for further processing. Another explanation may be that the regional differences in vascularization and local necrosis of tumor cells (which is typical for GBM) may facilitate a release of ccfDNA from specific regions.

Currently, detecting tumor-related genetic alterations in ccfDNA from a patient’s plasma is not sufficient to claim it as a diagnostic tool for glioma patients, but further research may lead to improvements in the procedure and better reproducibility. Altogether, we discovered potentially pathogenic variants in ccfDNA in 25 patients, including 24 glioma patients, although only in 5 cases were the same somatic SNVs consistently found in both ccfDNA and tumor DNA. We acknowledge that CSF in glioma patients offers better reliability and easier detection as the amount of brain-released ccfDNA is greater in CSF than in plasma, but the substantial invasiveness of the procedure for CSF biopsy must be considered. It is likely that the transient loosening of the blood–brain barrier, using mannitol or focused ultrasound blood–brain barrier disruption, can improve ctDNA detection rates. Moreover, our results indicated that a liquid biopsy from blood might yield a better representation of the overall spectrum of somatic variants present in the tumor, particularly in the case of metastatic patients. ccfDNA from blood likely contains tumor DNA from multiple tumor localizations, which can better assist in personalized therapy of primary and metastatic brain tumor patients [49,50,51,52]. A liquid biopsy from blood may be a plausible alternative for elderly brain tumor patients in which a biopsy of CSF is not recommended due to its invasiveness and danger of infection.

## 5. Conclusions

We demonstrated several technical improvements that allow for the precise control of the quality and quantity of ccfDNA and the application of targeted NGS to blood-derived ccfDNA. We detected ctDNA in 8 out of 84 patients, including 5 out of 80 glioma patients. Some ccfDNA showed somatic, pathogenic alterations that were not detectable in the matching tumor DNA. We concluded that while the sequencing of ccfDNA from blood has low efficacy, which prevents the use of this method as a diagnostic tool, further improvements to the isolation and processing of ccfDNA may make liquid biopsy also available for glioma patients. Our results show that liquid biopsy from blood is likely a better representation of the overall landscape of the somatic variants present in the tumor, particularly in the case of metastatic patients. ccfDNA from blood may contain tumor DNA from multiple tumor locations, which better reflects the genetic landscape of the primary and metastatic brain tumors [49,50,51,52]. Liquid biopsy from blood could also be an alternative for nonoperable brain tumors or elderly brain tumor patients in which a biopsy of CSF is not recommended due to its invasiveness.

## Figures and Tables

**Figure 1 cancers-14-03902-f001:**
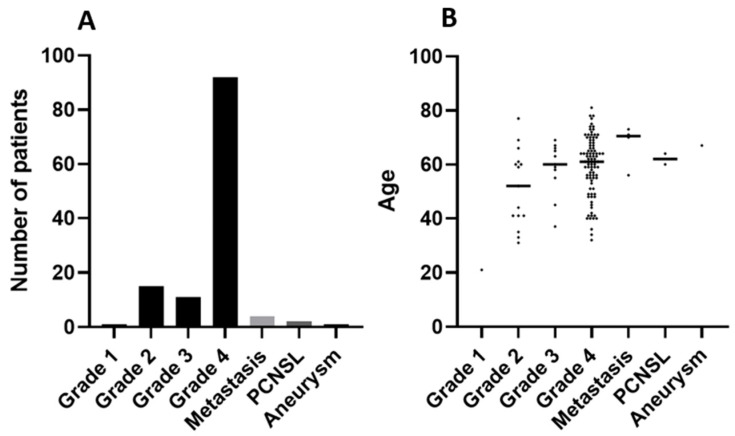
Summary of the patient cohort. (**A**) Numbers of specific tumor samples; (**B**) distribution of patients by age.

**Figure 2 cancers-14-03902-f002:**
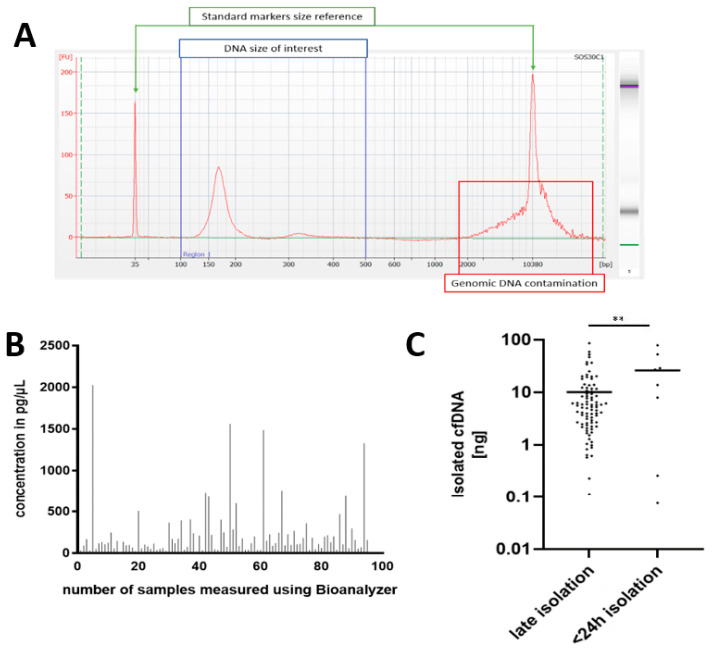
Quality and quantity control of ccfDNA samples evaluated using a Bioanalyzer. (**A**) Representative electropherogram: blue range highlights DNA size of interest for which concentration was determined; (**B**) concentration values for 95 ccfDNA samples measured by a Bioanalyzer; (**C**) effects of immediate versus delayed isolation on the yield of ccfDNA (blood that was left aside for more than 24 h after collection before isolation versus <24 h isolation). Statistical significance was calculated using a two-tailed *t*-test (** *p* < 0.01).

**Figure 3 cancers-14-03902-f003:**
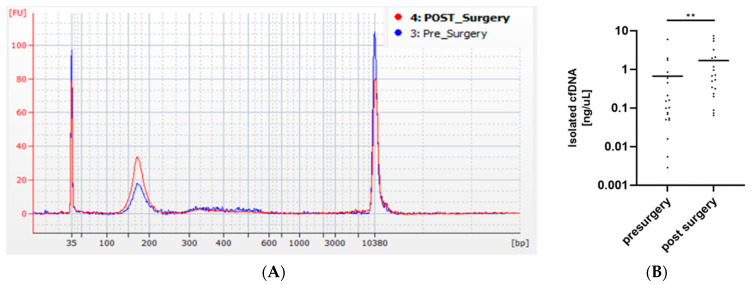
Comparison of yield of ccfDNA isolated from blood samples collected from the same patient before and after surgery. (**A**) Bioanalyzer electropherogram showing increased concentration in a sample of ccfDNA isolated from postsurgery (red) versus presurgery (blue); (**B**) comparison of total isolation yield shows significant increase in ccfDNA amounts isolated from postsurgical versus presurgery blood sample. Statistical significance was calculated using a paired *t*-test (** *p* < 0.01), N = 19.

**Figure 4 cancers-14-03902-f004:**
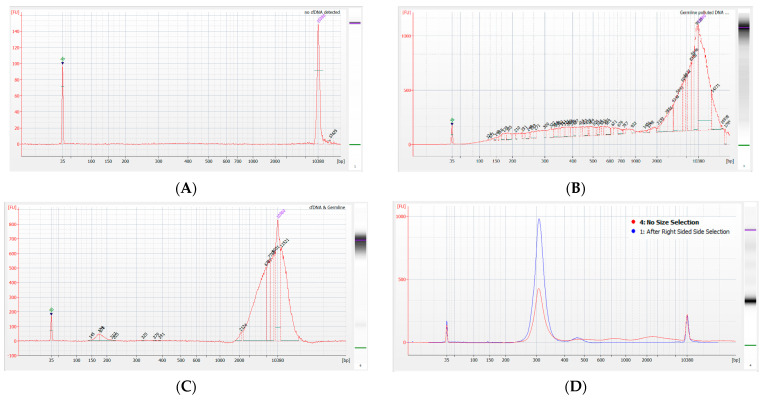
Bioanalyzer electropherograms illustrating quality control of ccfDNA after isolation and right-sided size selection. (**A**) ccfDNA sample containing no detectable material and excluded from the study; (**B**) ccfDNA sample that is strongly contaminated with fragmented genomic DNA and excluded from the study; (**C**) ccfDNA sample that contains some genomic DNA contamination and passed quality control; (**D**) ccfDNA derived libraries prior to hybridization and post first PCR, before (red) and after (blue) right-sided size selection.

**Figure 5 cancers-14-03902-f005:**
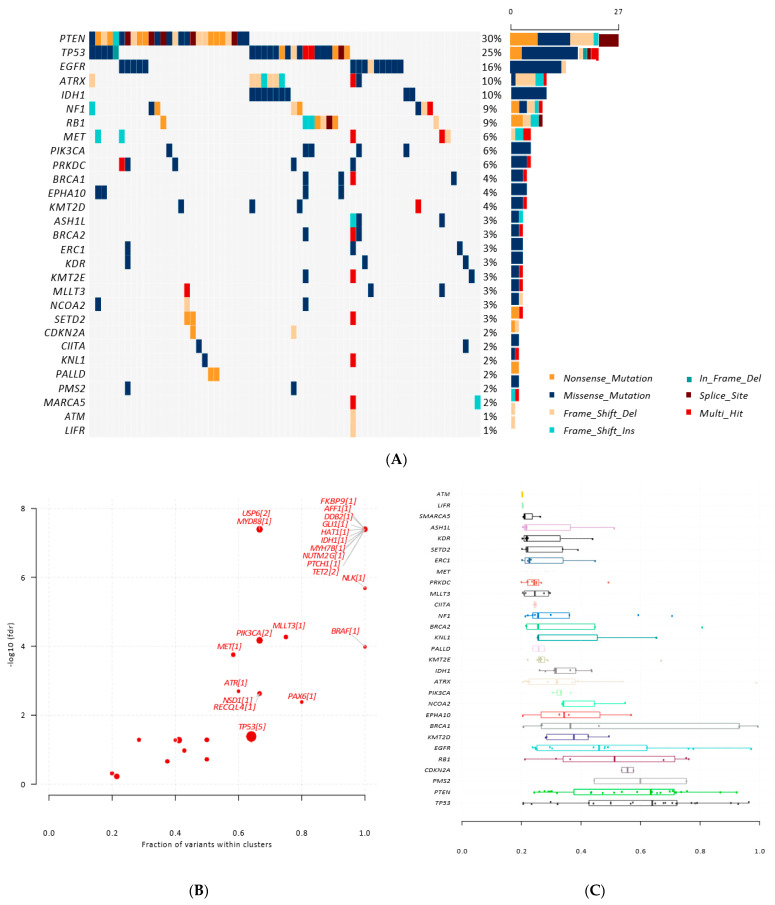
Somatic variants found in the tDNA: (**A**) mutational landscape plot of somatic variants found in the tumor tissue; (**B**) plot with size of the points proportional to a number of SNV clusters found in the gene. The *x*-axis indicates a fraction of the total variants identified in these clusters. Gene names are labeled along with the number of clusters found; (**C**) allele frequency variation between analyzed samples within a specific gene region.

**Figure 6 cancers-14-03902-f006:**
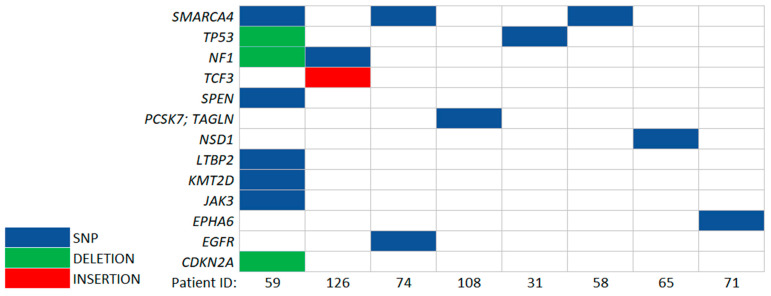
Somatic variants detected in ccfDNA and confirmed as somatic variants in tumor samples. The presence of specific alterations such as SNP, deletion, or insertion in a given patient sample is indicated.

**Figure 7 cancers-14-03902-f007:**
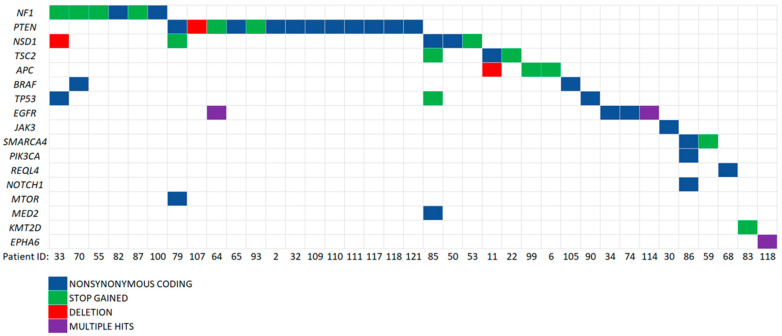
Detailed and schematic representation of genetic variants found in ccfDNA, but not in gDNA of brain tumor patients.

**Figure 8 cancers-14-03902-f008:**
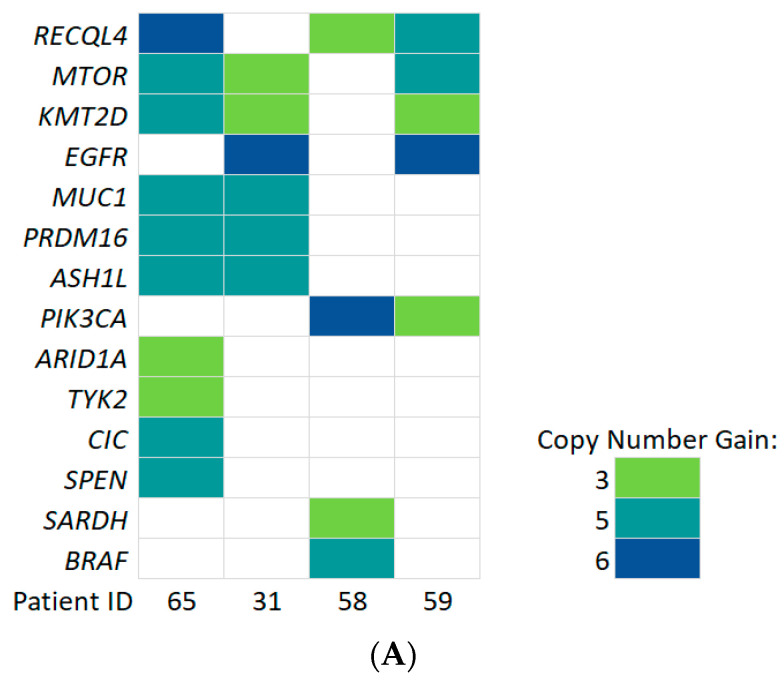
COSMIC registered copy number alterations detected in ccfDNA, but not gDNA of 4 brain tumor patients. (**A**) Graphical representation summarizing copy-number gain found in ccfDNA versus gDNA. Amplifications are shown in various colors to mark a copy number gains. (**B**) Detailed description of identified copy number gains.

**Table 1 cancers-14-03902-t001:** Somatic variants detected in ccfDNA and in the tumor sample.

							gDNA			Tumor DNA			ctDNA		
Gene	Chr	Position	ID	Ref	Alt	Diagnostic Information	Reads		AF	Reads		AF	Reads		AF
							All	Alt		All	Alt		All	Alt	
*TP53*	chr17	7577120	31	C	T	Glioblastoma, Grade 4	106	0	0	149	144	0.97	1219	11	0.009
*SMARCA4*	chr19	11170654	58	G	A	Primary Central Nervous System Lymphoma	49	0	0	55	23	0.42	602	9	0.015
*SMARCA4*	chr19	11144125	59	C	T	Anaplastic Thyroid Cancer Metastasis	203	0	0	77	55	0.71	1979	454	0.229
*TP53*	chr17	7579372		GC	G		324	0	0	128	76	0.59	1783	435	0.244
*SPEN*	chr1	16260997		G	T		237	0	0	141	82	0.58	2308	458	0.198
*KMT2D*	chr12	49438655		C	G		214	0	0	153	31	0.20	2238	229	0.102
*LTBP2*	chr14	75078119		T	G		20	0	0	15	9	0.60	279	50	0.179
*NF1*	chr17	29560103		GA	G		176	0	0	164	42	0.26	2444	227	0.093
*CDKN2A*	chr9	21971193		GC	G		148	0	0	95	66	0.69	1199	180	0.150
*JAK3*	chr19	17952151		G	T		29	0	0	9	5	0.56	1246	270	0.217
*NSD1*	chr5	176720936	65	G	C	Adenocarcinoma Lung Metastasis	390	0	0	418	191	0.46	862	183	0.212
*EPHA6*	chr3	96728829	71	G	GTT	Glioblastoma, Grade 4	11	0	0	23	3	0.13	618	14	0.023
*SMARCA4*	chr19	11144182	74	G	A	Astrocytoma Anaplasticum, Grade 3	43	0	0	259	76	0.29	1602	17	0.011
*EGFR*	chr7	55210075		T	G		123	0	0	3020	1514	0.50	1694	427	0.252
*PCSK7;TAGLN*	chr11	117076708	108	T	C	Glioblastoma, Grade 4	12	0	0	67	10	0.15	1539	340	0.221
*NF1*	chr17	29563087	126	T	G	Glioblastoma, Grade 4	67	0	0	112	3	0.03	2141	36	0.017
*TCF3*	chr19	1619749		A	AGGGTG		38	0	0	73	15	0.21	1281	310	0.242

**Table 2 cancers-14-03902-t002:** Cosmic registered variants found in ccfDNA but not in gDNA.

										gDNA (Maftools)			ccfDNA (SureCall)		
Gene	Chrom	Position	ID	rs ID	MAF	AF 1000 G	gnomAD	ClinVar clinsig	Diagnostic Information	Reads		AF	Reads		AF
										All	Alt		All	Alt	
*APC*	chr5	112177901	6	rs752654519	-	-	-	pathogenic/likely pathogenic	Glioblastoma, Grade 4	225	0	0	201	5	0.0249
*TSC2*	chr16	2098642	11	rs397515228	-	-	-	pathogenic	Diffuse Glioma, Grade 2	204	0	0	303	6	0.0198
*APC*	chr5	112111411		rs886039642	-	-	-	pathogenic/likely pathogenic		168	0	0	172	4	0.0233
*TSC2*	chr16	2136203	22	rs45517360	-	-	-	pathogenic	Glioblastoma, Grade 4	53	0	0	101	6	0.0594
*JAK3*	chr19	17950375	30	rs145751599	0	0.0004	2 × 10^−5^	uncertain significance	Glioblastoma, Grade 4	212	0	0	200	4	0.02
*NF1*	chr17	29677233	31	rs377662483	0	0.0002	2 × 10^−5^	uncertain significance	Glioblastoma, Grade 4	129	0	0	814	9	0.0111
*NF1*	chr17	29654553	33	rs876657714	-	-	-	pathogenic	Glioblastoma, Grade 4	252	0	0	470	4	0.00851
*TP53*	chr17	7577586		rs587781589	-	-	-	pathogenic		239	0	0	345	3	0.0087
*NSD1*	chr5	176637449		rs587784080	-	-	-	pathogenic		250	0	0	557	5	0.00898
*EGFR*	chr7	55233043	34	rs139236063	-	-	4 × 10^−6^	likely pathogenic	Glioblastoma, Grade 4	120	0	0	2897	39	0.0135
*NSD1*	chr5	176709524	50	rs587784169	-	-	-	pathogenic	Diffuse Astrocytoma, Grade 2	154	0	0	428	4	0.00935
*NSD1*	chr5	176696631	53	rs794727176	-	-	-	pathogenic	Glioblastoma, Grade 4	239	0	0	335	3	0.00896
*NF1*	chr17	29486070	55	rs746824139	-	-	0	pathogenic	Glioblastoma, Grade 4	144	0	0	424	5	0.0118
*PTEN*	chr10	89717695	64	rs190070312	-	-	-	pathogenic	Glioblastoma, Grade 4	246	0	0	436	5	0.0115
*PTEN*	chr10	89711900	65	rs121913294	-	-	-	likely pathogenic	Adenocarcinoma Lung Metastasis	139	0	0	352	3	0.00852
*RECQL4*	chr8	145741409	68	rs549497811	0	0.0002	2 × 10^−5^	uncertain significance	Glioblastoma, Grade 4	240	0	0	563	9	0.016
*BRAF*	chr7	140454008	70	rs397516894	-	-	-	pathogenic	Glioblastoma, Grade 4	228	0	0	372	4	0.0108
*NF1*	chr17	29562981		rs376576925	-	-	4 × 10^−6^	pathogenic		195	0	0	613	6	0.00979
*NF1*	chr17	29560088		rs878853884	-	-	-	pathogenic		118	0	0	626	8	0.0128
*MTOR*	chr1	11184573	79	rs587777894	-	-	-	pathogenic	Glioblastoma with Oligodendroglioma Component, Grade 4	133	0	0	250	8	0.032
*NSD1*	chr5	176673711		rs570278338	-	-	-	pathogenic		65	0	0	132	2	0.0152
*PTEN*	chr10	89692793		rs786204927	-	-	-	likely pathogenic		99	0	0	180	7	0.0389
*NF1*	chr17	29677228	82	rs533110479	0	0.0002	3 × 10^−5^	uncertain significance	Glioblastoma, Grade 4	243	0	0	436	5	0.0115
*KMT2D*	chr12	49438067	83	rs886043414	-	-	-	pathogenic	Glioblastoma, Grade 4	144	0	0	143	2	0.014
*TP53*	chr17	7579529	85	rs876658483	-	-	-	pathogenic	Glioblastoma, Grade 4	198	0	0	347	4	0.0115
*TSC2*	chr16	2114342		rs45517179	-	-	-	pathogenic		248	0	0	393	4	0.0102
*MED12*	chrX	70357138		rs762659794	0	0.0003	6 × 10^−6^	uncertain significance		115	0	0	161	4	0.0248
*PIK3CA*	chr3	178952085	86	rs121913279	-	-	4 × 10^−6^	pathogenic FDA recognized	Giant Cell Glioblastoma, Grade 4	245	0	0	139	2	0.0144
*NOTCH1*	chr9	139395108		rs371414501	0	0.0002	2 × 10^−5^	uncertain significance		178	0	0	165	3	0.0182
*SMARCA4*	chr19	11094931		rs563079629	0	0.0002	5 × 10^−5^	uncertain significance		58	0	0	136	5	0.0368
*NF1*	chr17	29588751	87	rs760703505	-	-	8 × 10^−6^	pathogenic/likely pathogenic	Glioblastoma, Grade 4	241	0	0	485	4	0.00825
*PTEN*	chr10	89720768	93	rs746930141	-	-	-	pathogenic	Glioblastoma, Grade 4	70	0	0	199	2	0.0101
*APC*	chr5	112173704	99	rs587779783	-	-	-	pathogenic	Diffuse Astrocytoma, Grade 2	250	0	0	643	6	0.00933
*NF1*	chr17	29490394	100	rs199474752	-	-	-	likely pathogenic	Glioblastoma, Grade 4	168	0	0	356	3	0.00843
*BRAF*	chr7	140453137	105	rs121913378	-	-	-	likely pathogenic	Pleomorphic Xanthoastrocytoma, Grade 2	209	0	0	273	8	0.0293
*PTEN*	chr10	89711968	107	rs587776670	-	-	-	pathogenic	Glioblastoma, Grade 4	104	0	0	482	4	0.0083

**Table 3 cancers-14-03902-t003:** Variants detected in ccfDNA, but not present in gDNA, confirmed as somatic in previous studies, tumor alterations.

							gDNA			ccfDNA			Tumor DNA
*Gene*	ID	rs ID	COSMIC (CNS)/Polyphen Pred	GMAF	ClinVar Clinsig	Diagnostic Information	Reads		AF	Reads		AF	AF (Somatic in Other Patient)
							All	Alt		All	Alt		
*PTEN*	2	rs12573787	-/-	0.16	benign	Oligoastrocytoma, Grade 2	60	0	0.00	255	8	0.031	0.6923
	32					Oligodendroglioma Anaplasticum, Grade 3	79	1	0.01	150	62	0.413	
	65					Adenocarcinoma Lung Metastasis	55	1	0.02	61	33	0.541	
	85					Glioblastoma, Grade 4	56	1	0.02	260	140	0.538	
	109					Anaplastic Pleomorphic Xantoastrocytoma, Grade 3	57	0	0.00	168	13	0.077	
	110					Glioblastoma, Grade 4	39	2	0.05	234	144	0.615	
	111					Glioblastoma, Grade 4	107	1	0.01	160	68	0.425	
	117					Glioblastoma, Grade 4	73	1	0.01	299	144	0.482	
	118					Glioblastoma, Grade 4	40	1	0.03	224	114	0.509	
	121					Giant Cell Glioblastoma, Grade 4	52	0	0.00	340	164	0.482	
*TP53*	90	rs121913343	131/D	-	pathogenic/likely pathogenic	Glioblastoma, Grade 4	147	3	0.02	1313	27	0.021	0.2619
*EGFR*	64	rs1057519828	14/D	-	likely pathogenic	Glioblastoma, Grade 4	225	0	0.00	612	22	0.036	0.4502
*EGFR*	64	rs149840192	36/D	-	likely pathogenic	Glioblastoma, Grade 4	181	3	0.02	703	9	0.013	0.248

**Table 4 cancers-14-03902-t004:** Variants detected in ccfDNA, but not present in gDNA, confirmed as somatic in current study, tumor alterations.

										gDNA (Maftools)			ccfDNA (SureCall)			Tumor
*Gene*	Chr	Position	Ref	Alt	ID	rs ID	Registered in COSMIC	ClinVar Clinsig	Diagnostic Information	Reads		AF	Reads		AF	# of Patients with Somatic Variant
										All	Alt		All	Alt		
*SMARCA4*	19	11144125	C	T	59		yes	-	Anaplastic Thyroid Cancer Metastasis	186	0	0	1979	454	0.229409	1
*PIK3CA*	3	178952085	A	G	86	rs121913279	-	likely pathogenic	Giant Cell Glioblastoma, Grade 4	245	0	0	139	2	0.014388	1
*EPHA6*	3	97365038	G	A	118	rs301948	-	-	Glioblastoma, Grade 4	199	0	0	1328	86	0.064759	1
*EPHA6*	3	97365074	A	G	118	rs301949	yes	-	Glioblastoma, Grade 4	179	0	0	1611	103	0.063935	1
*EGFR*	7	55210075	T	G	74		yes	-	Astrocytoma Anaplasticum, Grade 3	229	0	0	1694	427	0.252066	2
*EGFR*	7	55210075	T	G	114		yes	-	Astrocytoma, Grade 3	247	1	0.004	274	3	0.010949	2
*EGFR*	7	55224307	C	T	114		yes	likely pathogenic	Astrocytoma, Grade 3	245	0	0	417	9	0.021583	1
*EGFR*	7	55221822	C	T	64	rs149840192	yes	-	Glioblastoma, Grade 4	181	3	0.0166	703	9	0.012802	3

## Data Availability

The data from this study are openly available at the European Genome-phenome Archive (EGA), reference number EGAS00001006451 and EGAD00001009080.

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
