# Peer review of "Improvements in Quality Control and Library Preparation for Targeted Sequencing Allowed Detection of Potentially Pathogenic Alterations in Circulating Cell-Free DNA Derived from Plasma of Brain Tumor Patients"

_cancers, 2022, doi:10.3390/cancers14163902_

Round 1

Reviewer 1 Report

This study is undoubtedly relevant and represents a certain interest for molecular biologists and oncologists. However, have some questions for authors:

1.   In my opinion, the authors offer not quite the correct title of the article.

2.   At what time after surgery blood samples were carried out in order to isolate cirDNA? This information not indicated in article.

It is known that immediately after the surgery the cirDNA concentration can sharply increase and it is not entirely appropriate to carry out blood sample collecting during this period.

3.   …”Unique genetic variants found in cfDNA, but not in reference DNA…” These variants detected in content of pre- and post-surgery isolated cfDNA? These findings will be validated on independent cohort?

Author Response

We would like to thank the reviewer for the questions and comments. Suggested improvements and clarifications are addressed below and included in the revised manuscript.

  1. In my opinion, the authors offer not quite the correct title of the article.

We thank the reviewer for a suggestion and we adjusted the title: “Improvements in quality control and library preparation for targeted sequencing allowed detection of potentially pathogenic alterations in circulating cell-free DNA derived from plasma of brain tumor patients”. We believe the modified title represents the content fittingly.

  1. At what time after surgery blood samples were carried out in order to isolate cirDNA? This information not indicated in article.

Directly after the surgery the ccfDNA concentration can sharply increase and we did not collect blood sample during this period. Most of the samples were collected 2-3 days post-surgery, for the rest 4-5 days. We included this information in the revised ,manuscript methods sections, line 134-135: “Post-surgical blood for ccfDNA isolation was collected from most patients 2-3 days after surgery, in some cases 4-5 days post-operation.”

  1. …”Unique genetic variants found in cfDNA, but not in reference DNA…” These variants detected in content of pre- and post-surgery isolated cfDNA? These findings will be validated on independent cohort?

Usage of the word “unique” referred to the detected somatic variants that were only present in ccfDNA, but not in gDNA. We agree that wording might not be precise and could be misleading, so we removed this word at this specific context from the manuscript.

All ccfDNA sequencing was performed on pre-surgery isolated ccfDNA. The information is included in the revised manuscript in Methods section, line 121-126: “In the NGS analysis we used pre-surgery plasma derived ccfDNA collected from 84 patients: 80 patients with WHO G3 and G4 gliomas, 2 patients with Primary Central Nervous System Lymphoma (PCNSL), and 2 patients with anaplastic thyroid cancer metastasis and adenocarcinoma lung metastasis. Additionally, copy number alteration targeted sequencing was done on 4 ccfDNA samples that displayed ctDNA signal in the primary analysis”.

Reviewer 2 Report

The manuscript by Szadkowska and colleagues describes several strategies to improve variants detection, including cfDNA isolation, library preparation and NGS data bioinformatics, with a cohort of 84 paired samples of tumor and blood (before and after surgery). 

Overall, the cohort of patients are of high interests, and the design of experiments are property. The assays used in the data generation is well accepted. But the writing and data presentation could be improved.

 Issues to be considered:

1. Serum-derived cfDNA vs plasma-derived cfDNA? How the serum/plasma was prepared should be clearly addressed in method part. How many rounds of centrifugation with what speed?

2.    In the title, “specific alterations”? How are these alterations specific? Disease specific or sample type specific? Should be clarify.

3.    Grammar mistake throughout the manuscript….i.e., P1 Line 21,22

4.    Two compartments of blood were analyzed in the study. Please specify gDNA and cfDNA in the text, table, and figures. Do not use “blood DNA”.

5.    Redundant key words, circulating tumor DNA vs ctDNA???

6.    Missing figure legend for the graphic abstract

7. Since different capture assay was used for tumor/gDNA and cfDNA samples, it will be great to have a summary graph to indicate the overlapping of capture regions. 50gene of SureSelect all covered by SeqCap 578 genes?

8.    Why use different ref genome for bioinformatics?

9. Would different NGS setting affect data analysis? Why use 2X76 for tumor/gDNA, but 2X100 for cfDNA?

10. Most commercial Library construction kits do not recommend fragmentation of cfDNA samples. Is it necessary to be pointed out as an innovative improvement?

11. Size selection direction left or right? P14 Line 453 says left side, while figure shows right sided size selection.

12. Size selection BioA profile in Figure 4, what are the PCR cycles for these two samples? Is this peak increasing due to more PCR cycles?

13. Somatic CNA detection: in method part, mentioned CNA was calculated in tumor and blood/tumor pair, while in results table 5 and figure 8 it is cfDNA. Table 5 and Figure 8 could be merged into one Figure.

14. Overall, redundant table and images. Most of results shown with one table + one Figure. Could be merged and simplified for data visualization

Author Response

We would like to thank the Reviewer for many questions and comments. Suggested improvements and clarifications are addressed below. Implemented changes are included in the revised manuscript.

  1. Serum-derived cfDNA vs plasma-derived cfDNA? How the serum/plasma was prepared should be clearly addressed in method part. How many rounds of centrifugation with what speed?

Thank you for this comment. We work on plasma derived ccfDNA. We included the additional information describing all steps of plasma processing in the manuscript. Additional paragraph with detailed description of cfDNA isolation was added in the revised text, line 138-147.

  1. In the title, “specific alterations”? How are these alterations specific? Disease specific or sample type specific? Should be clarify.

Usage of word “specific” might have been misleading. This description was supposed to refer to non-germline, potentially pathogenic variants that are detectable in ccfDNA, but not in gDNA. These alterations are meant to be “tumor specific”. The title was changed: Improvements in quality control and library preparation for targeted sequencing allowed detection of potentially pathogenic alterations in circulating cell-free DNA derived from plasma of brain tumor patients”. We believe the modified title represents the content fittingly.

  1. Grammar mistake throughout the manuscript….i.e., P1 Line 21,22

Line 21-22 has been corrected and the entire text was corrected as is shown in the tracked changes version.

  1. Two compartments of blood were analyzed in the study. Please specify gDNA and cfDNA in the text, table, and figures. Do not use “blood DNA”.

We will include term gDNA for germline whole blood reference DNA. We also introduced tDNA for tumor derived DNA. We hope this will make the text more clear and easier to read. We agree that using blood DNA was unfortunate.

  1. Redundant key words, circulating tumor DNA vs ctDNA???

We introduced proper abbreviations: ccfDNA, ctDNA, tDNA and gDNA and corrected key words to avoid repetition.

  1. Missing figure legend for the graphic abstract

In Cancers a legend for the graphic abstract is not required.  Abstract was designed to be self-explanatory.

  1. Since different capture assay was used for tumor/gDNA and cfDNA samples, it will be great to have a summary graph to indicate the overlapping of capture regions. 50gene of SureSelect all covered by SeqCap 578 genes?

All targeted area of interest were overlapping. The SeqCap panel contains 664 gene regions: including 578 selected from the Roche Nimblegen Cancer Comprehensive Panel and additional 86 epigenetics-related genes (see the revised text). All 50 genes of SureSelect are included in the larger panel and this information is highlighted in the revised text, line 164-165: “All selected gene regions were included in the 664 gene larger panel (Supplementary File S2)”. Instead of a summary graph, we provided information about gene panels in the Supplementary Bed Files: S1 (664 gene panel), S2 (50 gene panel), S3 (50 gene CNA panel). 

  1. Why use different ref genome for bioinformatics?

At the time we performed the experiments, the Agilent SureSelect XT was the only custom panel design library preparation kit that contained UMI (unique molecular identifier) and it’s manufacturer recommended analysis software that generated vcf in old ref genome format. Agilent analysis pipeline is a proprietary software solution, that we couldn’t modify. This is a main reason why there are two different ref genome for SeqCap (664 gene panel) and Sure Call (50 gene panel). In our opinion it is beneficial to use hg38 genome annotation, so that is the reference genome that we have used in our pipeline analysis.

  1. Would different NGS setting affect data analysis? Why use 2X76 for tumor/gDNA, but 2X100 for cfDNA?

We followed the recommendations of the company providing the kit and reagents. 2x100 setting of the cfDNA library sequencing was used as recommended by Agilent. We did not want to modify this setting, because in case of any claims and reclamations it is the best to stick exactly to producer defined settings. Tumor DNA and gDNA was sequenced 2x76, as this was the optimized setting for our established internal bioinformatics analysis pipeline, provided sufficient sequencing depth and lowered sequencing costs. This NGS setting is used according to general recommendations. Indeed, when library size is around 320 bp, 2x76 sequencing setting is optimal, not to read library insert size twice, both from one and 2nd end of a library fragment.

Different settings 2x100 vs 2x76 will not change final results as both settings are very well mappable to the genome. The computational analytical pipeline was similar. Indeed, ccfDNA was not only sequenced 2x100, but also ultra-deep to find low-frequency variants, as suggested by the producer.

  1. Most commercial Library construction kits do not recommend fragmentation of cfDNA samples. Is it necessary to be pointed out as an innovative improvement?

We meant that standard NGS sequencing protocols (including ours) encompass DNA fragmentation using Covaris. We found it worthy to mention, as it is crucial not to fragment ccfDNA prior to library preparation. However, we agree with the reviewer that this is not worth emphasizing, as it is a common knowledge by now.

  1. Size selection direction left or right? P14 Line 453 says left side, while figure shows right sided size selection.

Right sided size selection was used in order to enrich the cfDNA libraries into shorter fragments. The mistake is fixed. Thank you for this important comment.

  1. Size selection BioA profile in Figure 4, what are the PCR cycles for these two samples? Is this peak increasing due to more PCR cycles?

Our comment was misleading. Fig. 4D illustrates the same library, after the same number of PCR cycles (14), but before and after performing additional step of right sided size selection. It shows that there is the enrichment into shorter fragments of the prepared library after size selection. Figure description was revised to illustrate that better.

  1. Somatic CNA detection: in method part, mentioned CNA was calculated in tumor and blood/tumor pair, while in results table 5 and figure 8 it is cfDNA. Table 5 and Figure 8 could be merged into one Figure.

CNA was analyzed by comparing ccfDNA and gDNA, by use of the analysis software (Surecall) recommended by producer. We have used only those ccfDNA samples in which the same somatic alterations were identified in ccfDNA and tDNA. Amplifications identified only in ccfDNA, but not in gDNA that are also registered in COSMIC, are presented. We agree that one figure can summarize these information, so we merged table 5 and figure 8 as one figure.

  1. Overall, redundant table and images. Most of results shown with one table + one Figure. Could be merged and simplified for data visualization

Thank you for your suggestion, it was done for the figure 8. The information in the tables and figure panels is not redundant, as a table contains more details about genetic alterations (chromosome, coordinates, exact base substitution information, etc.) and the figure is mostly a general graphical representation. We could not find a better way to present all of the details of our data. For the rest of the tables we just cohered to the editor’s suggestion to change the table format to MS Word and kept the tables. We hope this makes the text easier to follow.

We attached "track changes" version of the manuscript, so the extent of implemented changes can be visible to the Reviewer (author-coverletter-20904532.v1.dox).

Round 2

Reviewer 2 Report

Great job of revising.